# Fault Diagnosis of Planetary Gearbox Based on Adaptive Order Bispectrum Slice and Fault Characteristics Energy Ratio Analysis

**DOI:** 10.3390/s20082433

**Published:** 2020-04-24

**Authors:** Zhaoyang Shen, Zhanqun Shi, Dong Zhen, Hao Zhang, Fengshou Gu

**Affiliations:** 1Tianjin Key Laboratory of Power Transmission and Safety Technology for New Energy Vehicles, School of Mechanical Engineering, Hebei University of Technology, Tianjin 300401, China; 201711201015@stu.hebut.edu.cn (Z.S.); d.zhen@hebut.edu.cn (D.Z.); zhanghao@hebut.edu.cn (H.Z.); 2Centre for Efficiency and Performance Engineering, University of Huddersfield, Huddersfield HD1 3DH, UK; f.gu@hud.ac.uk

**Keywords:** fault diagnosis, adaptive order bispectrum slice, fault characteristics energy ratio, planetary gearbox

## Abstract

The vibration of a planetary gearbox (PG) is complex and mutually modulated, which makes the weak features of incipient fault difficult to detect. To target this problem, a novel method, based on an adaptive order bispectrum slice (AOBS) and the fault characteristics energy ratio (FCER), is proposed. The order bispectrum (OB) method has shown its effectiveness in the feature extraction of bearings and fixed-shaft gearboxes. However, the effectiveness of the PG still needs to be explored. The FCER is developed to sum up the fault information, which is scattered by mutual modulation. In this method, the raw vibration signal is firstly converted to that in the angle domain. Secondly, the characteristic slice of AOBS is extracted. Different from the conventional OB method, the AOBS is extracted by searching for a characteristic carrier frequency adaptively in the sensitive range of signal coupling. Finally, the FCER is summed up and calculated from the fault features that were dispersed in the characteristic slice. Experimental data was processed, using both the AOBS-FCER method, and the method that combines order spectrum analysis with sideband energy ratio (OSA-SER), respectively. Results indicated that the new method is effective in incipient fault feature extraction, compared with the methods of OB and OSA-SER.

## 1. Introduction

The planetary gearbox (PG) is widely used in robots, wind turbines, ships, heavy equipment and helicopters [1,2]. Condition monitoring and fault-diagnosis for the PG are of great significance to production safety and cost efficiency [3]. Fault signatures of the PG can manifest in vibrations, acoustic and electrical signals, temperature, and oil-debris characteristics. Vibration measurement is one of the most widely-used approaches to monitor and diagnose the health of a PG [4]. In a PG, planetary gears rotate not only on themselves, but around the sun gear at the same time [5]. This rotation manner gives rise to a unique, complex and mutually modulated performance in the vibration signal of the PG. Even under normal conditions, there are many sidebands in the spectrum of the vibration signal of a PG [6]. Therefore, fault-diagnosis for the PG is more difficult than it is for the fixed-shaft gearbox [7]. It is well known that the earlier the fault is diagnosed, the lower the loss. However, the incipient fault characteristics of the PG are weak, and modulated in complex vibration. Therefore, how to extract the incipient fault feature of a PG has become a challenging problem.

In recent years, various advanced signal processing approaches have been used to extract fault features from the vibration signal of a PG [8,9,10,11,12,13,14]. For instance, G. He et al. exploited a two-dimensional, centralized model for the PG, with a floating sun gear [8]. Z. Feng et al. decomposed the vibration signal, using the intrinsic time-scale method and the local mean decomposition method separately to get the sensitive component fluctuating around the meshing frequency—that is, its harmonics—and diagnosed the fault by analyzing the amplitude and frequency of the sensitive component [9,10]. Y. Li et al. developed an improved Vold–Kalman filter and a multi-scale sample entropy method, to extract the PG’s fault features under non-stationary working conditions [11]. T. Wang et al. improved the feature extraction capacity of spectral kurtosis, which uses the time wavelet energy spectrum, and an adaptive, tunable Q-factor wavelet transform filter algorithm [12,13]. J. Guo et al. employed the wavelet packet energy method and Modulation Signal Bispectrum (MSB) method to detect incipient faults in a PG [14]. The above approaches have advanced our capacity to extract fault features from PGs, but are mainly based on nonlinear component demodulation, and ignore the nonlinear effect caused by speed fluctuation. Therefore, these methods will inevitably induce pollutants when extracting the weak fault features of a PG.

In order to detect the faults of a PG accurately, two problems need to be solved:

(1)The modulation sidebands exist inherently as long as the sun gear is floating [8]. This implies that the conventional fault features can also be extracted from a healthy PG. Therefore, the conventional features fail to distinguish between health and faultiness in the PG.(2)The vibration behavior of a PG is unique to the transmission path [15,16]. This means the features extracted via conventional methods vary under different sampling.

The order bispectrum (OB) is an excellent nonlinear feature extraction method, which can reduce the impact of speed fluctuation and is widely used in the field of fault diagnosis [17,18,19,20,21]. It was introduced, by D. Kocur et al., to analyse the vibration of reciprocating machines [17]. J. Han confirmed that OB is more effective than bispectrum in extracting rotor imbalance fault [18]. H. Li used the OB method to diagnose faults on bearings under speed run-up conditions [19]. Z. Liu extracted bearing faults, caused by vibration, using linear frequency modulation wavelet path tracking and the OB method [20]. Y. Zhang used OB to diagnose the fault of gear crack in the fixed-shaft gearbox under speed run-up conditions [21]. Although the application of the OB method has been mentioned in the context of fault diagnosis in PGs, very few cases can be found in the existing literature.

Inspired by Ref. [19], the characteristic slice of OB for the PG can be found by searching a sensitive peak cluster adaptively, giving the AOBS that is proposed in this paper. The sideband energy ratio (SER) is used to highlight the sideband information [22]. It represents the energy ratio of the sum of the different order harmonics’ sidebands to the meshing frequency [23]. Enlightened by Refs. [22,23], the ratio of the amplitudes of the homologous modulation peak to the central frequency can create a new feature, via weighted combination, which reduces the effect of the variation of transmission path on the feature value. Thus, a novel method based on AOBS and FCER is proposed in this paper. The AOBS method can extract weak fault features in the low frequency range, where the complexities of the modulation component are lesser. The FCER can smooth the variations of the conventional features. To validate the effectiveness of the AOBS-FCER method, experiments have been designed focusing on incipient faults in a PG system versus a healthy PG system, under different speed and load conditions. A comprehensive analysis of the AOBS-FCER method has been performed, and it has been compared with the methods of OB and sideband energy ratio analysis based on order spectrum analysis (OSA-SER) in the same experimental dataset. The results validate that the proposed method is effective and superior in identifying the weak faults in a PG.

The rest of the paper is organized as follows: Section 2 describes the fundamentals of the proposed AOBS-FCER method; Section 3 illustrates the specific steps of the AOBS-FCER method; Section 4 concerns experiments conducted to verify the effectiveness of the AOBS-FCER method, and a comparison of the results, which validate the superiority of the new method. Finally, the conclusions of this paper are drawn in Section 5.

## 2. Fundamentals of the Proposed AOBS-FCER Method

In this section, the fundamentals of the AOBS-FCER method are described. The AOBS is described in two subsections concerning the order spectrum and the adaptive bispectrum slice, respectively. The FCER is briefly introduced, and its effectiveness is validated by comparison with the SER via numerical simulation.

### 2.1. Fault Characteristic Detection of Adaptive Order Bispectrum Slice

#### 2.1.1. The Angle-Domain Synchronous Average and Order Spectrum

Order spectrum analysis (OSA) is a useful tool for reducing the impact of speed fluctuation on rotating machinery, and is widely used in the fault diagnosis of PGs [24,25,26,27,28]. The basic element of OSA is the angle-domain synchronous average (ADSA). Since the duration of each revolution of the rotating machinery is unequal, owing to the speed fluctuation, the vibration signal sampled in the time domain is non-stationary. However, the angle of each revolution of rotating machinery is stationary. Therefore, vibration in the angular domain can be regarded as stationary, and the influence of speed fluctuation can be depressed.

The first step in deducing the ADSA is to collect the encoder signal from the sun gear shaft and the vibration signal from the gear ring of the PG synchronously. Next, the vibration signal is resampled at an equal angle. Since the rotation speed can be described as uniform acceleration motion in a short time, this process can be expressed by Equation (1).
(1)$$\theta (t)=a_{0}+a_{1}\cdot t+a_{2}\cdot t^{2}$$
where θ(t) is the angle that corresponds to the pulse acquired by the encoder. $a_{0}$, $a_{1}$ and $a_{2}$ are the undetermined coefficients, and they can be calculated by putting the collected angles of the integer circles of the reference shaft into Equation (2).
(2)$$\left(\begin{array}{c} \theta (t_{0})\\ \theta (t_{1})\\ \theta (t_{2}) \end{array}\right)=\left(\begin{array}{ccc} 1 & t_{0} & t_{0}{}^{2}\\ 1 & t_{1} & t_{1}{}^{2}\\ 1 & t_{2} & t_{2}{}^{2} \end{array}\right)\cdot \left(\begin{array}{c} a_{0}\\ a_{1}\\ a_{2} \end{array}\right)$$
where the initial values of θ(t) are 2π, 4π and 6π (the first pulse from the encoder is regarded as the angle of 0), $t_{0}$, $t_{1}$ and $t_{2}$ are their corresponding sampling times, respectively. Thus, $a_{0}$, $a_{1}$ and $a_{2}$, in the range from $\theta (t_{0})$ to $\theta (t_{2})$, are obtained.

The discrete resampling points with equal angle intervals can then be calculated by Equation (3).
(3)$$t=\frac{1}{2a}\left[\sqrt{4a_{2}(k\cdot \Delta \theta -a_{0})+a_{1}^{2}}-a_{1}\right]$$
where t represents the sampling time of the equal angle interval in the time domain, k is the integer, and Δθ denotes the equal angle interval. The vibration signal in the angular domain can be obtained by interpolating t in the raw vibration signal. The sampling frequency in the angular domain is 2π/Δθ.

The OSA transforms the vibration signal in the angular domain using the Discrete Fourier transform (DFT) [29]. Since the revolution period is the same as the encoder, the order spectrum of the reference shaft is 1 Order. The other frequencies can be obtained by calculating the proportions of their frequencies w to the sun gear rotation frequency $w_{s}$. Therefore, the “sun gear order” can be the unit in the OSA. Thus, the signal processing in the frequency domain is no longer related to the speed, and the nonlinearity caused by speed fluctuation is depressed.

#### 2.1.2. The Bispectrum and Adaptive Bispectrum Slice

Compared with traditional spectrum analysis, the results of higher order spectrum analysis contain not only the deviation degrees from Gaussian distribution, but also the nonlinear existence information of signal [30]. The high order spectrum is based on the high order cumulant, and is therefore also called the high order cumulant spectrum [31]. Bispectrum, the simplest and most widely-used high order spectrum, is the multiple Fourier transformation of the third order cumulant. There are two important parameters in bispectrum calculation, which are carrier frequency and modulation frequency.

The vibration of the PG with a floating sun gear has complex modulation sidebands [32]. Suppose the center frequency of a double-sideband is $f_{0}$, the sidebands are $f_{1}$ and $f_{2}$, and they have the behavior of $\left|f_{1}-f_{0}\right|=\left|f_{2}-f_{0}\right|$, otherwise, $\left|f_{1}-f_{0}\right|\neq \left|f_{2}-f_{0}\right|$. Then $f_{0}$ is the carrier frequency, and the interval between the sideband and the center frequency is the modulation frequency. The bispectrum of the energy finite signal, both discrete and stationary {s(n),n=0,±1,±2,⋯}, can be computed under the sum of its absolute values of the k-order cumulants of its finite, which can be expressed as Equation (4) [33,34,35,36].
(4)$$B_{s}(w_{1},w_{2})=E\left[S(w_{1})\cdot S(w_{2})\cdot S^{*}(w_{1}+w_{2})\right]$$
where $w_{1}$ and $w_{2}$ are the carrier frequency and modulation frequency, respectively. The S(w) represents the Fourier transform of s(n), $S^{*}(w)$ is the conjugate complex function of S(w), and E[•] denotes the mathematical expectation function of inner product space.

Since the inner products have useful values only in the case of nonlinear coupling, the bispectrum can effectively describe and reflect the characteristics of nonlinear components containing the fault frequencies. Similar to the relationship of correlation function and power spectrum for the stationary signal, the Fourier transform relationship exists between the high order spectrum of stochastic processes and their corresponding cumulants. Suppose that x(t),(t=1,2,3,⋯) is a real stationary discrete stochastic process, which is finite and has a mean of zero. Its third order cumulant can be defined as Equation (5).
(5)$$c_{3}(m,n)=\frac{1}{L}x(t)\cdot x(t+m)\cdot x(t+n)$$
where L is the length of the series, and m and n are two independent parameters, which denote time delay.

The bispectrum values on the flat surface of $\left(w_{1},w_{2}\right)$ can then be worked out with the two-dimensional discrete Fourier transform (DFT), as shown by Equation (6).
(6)$$B_{x}(w_{1},w_{2})=\sum_{m=-L_{1}}^{L_{1}}\sum_{n=-L_{2}}^{L_{2}}c_{3}(m,n)e^{j(w_{1}m+w_{2}n)}$$
where the lengths of $L_{1}$ and $L_{2}$ determine the capacity of the bispectrum.

The formation of bispectral peaks on the flat surface of $\left(w_{1},w_{2}\right)$ is shown in Figure 1. Suppose that $w_{20}$, $w_{21}$ and $w_{22}$ are nonlinear with $w_{10}$, $w_{11}$, and $w_{12}$, respectively. Then the peaks will appear at the points of $\left(w_{10},w_{20}\right)$, $\left(w_{10},w_{21}\right)$, $\left(w_{10},w_{22}\right)$, $\left(w_{11},w_{20}\right)$, $\left(w_{11},w_{21}\right)$, $\left(w_{11},w_{22}\right)$, $\left(w_{12},w_{20}\right)$, $\left(w_{12},w_{21}\right)$ and $\left(w_{12},w_{22}\right)$. With the increase of carrier frequency, the modulation components become more and more complex. The slice of bispectrum can highlight the low frequency spectral line. A conventional slice spectrum is a diagonal slice, horizontal slice and vertical slice [13,14,15,16]. However, the bispectral peak value is effected by the degree of nonlinearity between each $w_{1}$ component and $w_{2}$ component. Therefore, the adaptive bispectrum slice is exploited to extract a characteristic slice of bispectrum on the characteristic carrier frequency adaptively. The sideband peaks of a same carrier frequency can be summed up to enhance the energy of the carrier frequency, based on Ref. [21]. Similarly, the modulation components on the same carrier frequency of the higher order spectrum can also be added to highlight the carrier frequency, as shown in Equation (7). Then the characteristic slice of AOBS can be extracted, as in Equation (8).
(7)$$B_{xs}=\sum_{w_{1}}B_{x}(w_{1},w_{2})$$
(8)$$B_{xs}=\max\left\{B_{xs}(w_{2})\right\}$$

### 2.2. Fault Symptoms Representation of Fault Characteristic Energy Ratio

Since the vibration components in a PG are mutually modulated, the characteristics extracted in an OB slice are various. Moreover, these characteristic values vary with the sampling time, owing to the time-varying transmission path of vibration in a PG. These phenomena make it difficult to accurately infer the existence of faults from the extracted characteristics. However, the power of the impact caused by the fault is stable in the extracted OB slice with the sampling time. According to Ref. [4], each order of the harmonic frequency of the sun gear exists while it is floating. The total energy created by the meshing between the sun gear and the planetary gear is constant [5]. In order to simplify the calculation, the vibration of the sun gear can be supposed as Equation (10).
(9)$$x_{sun}(t)=\sum_{i=1}^{N}a_{i}\cos(2\pi if_{s}t),(i=1,2,3,\cdots )$$
where $a_{i}$ denotes the weight of each order of the harmonic frequency of the sun gear, it is variable and $\sum_{N}a_{i}=1$.

Thus the fault characteristics that originate in the same component of the PG are summed up on the extracted OB slice. Inspired by Ref. [22,23], the fault characteristics that originate in the same component of the PG can be summed up and calculated on the AOBS. Now, a new feature of FCER, for representing the proportion of the overall energy of a rotating component that is abnormal energy, is proposed in this paper. The statistical fault signature of FCER can be calculated as Equation (9).
(10)$$FCER=\ln\left|\sum_{i=1}^{k}A_{fi}/E(A_{c})\right|$$
where $A_{f}$ is the characteristic values extracted by the OB of different harmonic frequencies originating in the same rotating component in a PG, $A_{fi}$ represents the amplitude of the ith order harmonic nonlinear phase coupling characteristic, k is the number of harmonics of $A_{f}$, and $E(A_{c})$ denotes the expected amplitude of the modulation frequencies on the characteristic slice.

To verify the identification performance of the FCER, the FCER is compared with the SER via the numerical simulation signal in Equation (11).
(11)$$x(t)=\cos(2\pi f_{1}t)+\sum_{i=1}^{4}a_{i}\cdot \cos[2\pi (f_{1}\pm i\cdot f_{2})t]+3randn(t)$$
where $f_{1}$ is the carrier frequency, and $f_{1}=50$ Hz. $f_{2}$ is the modulation frequency, and the $f_{2}=13$ Hz. The weight $a_{i}$ of harmonics of the modulation frequency $f_{2}$ is simulated randomly. Four values of $a_{i}$ in the simulation group are shown in Table 1. The signal x(t) is sampled with a sampling frequency of 1024 Hz and 4096 samples.

The waveforms of the simulated signals are depicted in Figure 2. The frequency and extracted characteristic slice of the simulated signal are shown in Figure 3 and Figure 4, respectively. The SER and FCER are then applied to process the simulated signal in Figure 1, and the results are shown in Table 2. As can be seen from Table 2, the FCER has better convergence than the SER. It indicates that a threshold range can be used in the FCER method to represent the abnormal characteristics. The results show that the FCER is more suitable for representing the weak fault features of a PG.

## 3. Application of the AOBS-FCER Method

Based on the advantages of AOBS and FCER, the AOBS-FCER method is proposed for fault diagnosis in the planetary gearbox. Four steps are required in the application of the proposed AOBS-FCER method, as follows:Convert the vibration signal in the time domain to that in the angular domain by the ADSA method.Calculate the OB. The frequency ranges of the modulation components and carrier components affect the OB values; in particular, the carrier frequency has a great influence on the results, and a wide range of it should be selected whenever possible.Extract the characteristic slice by Equation (8).Sum up and calculate the FCER.The flowchart of the AOBS-FCER method is drawn in Figure 5.

## 4. Experimental Validation

In this section, a pitting fault in a sun gear was created to represent the incipient fault in a PG. Experiments of the sun gear pitting fault PG versus the healthy PG, under different load and speed conditions, were performed to validate the effectiveness of the proposed AOBS-FCER method.

### 4.1. Experiment Setup

The experimental system is composed of a three-phase AC motor, a two-stage helical gearbox, a PG and a DC generator, as shown in Figure 6. The system is driven by the AC motor, the rated power and rated speed of which are 11 KW and 1500 rpm, respectively. The two-stage helical gearbox is directly connected to the motor output. Its first stage is used to decelerate and the second stage is used to accelerate, and the contact ratios of each are 1.45 and 1.469, respectively. The PG consists of a sun gear, three planet gears and a fixed internal ring gear. Its input shaft and output shaft are connected with the helical gearbox and the DC generator, respectively, through the flexible coupling. The configuration parameters for the planetary gearbox are listed in Table 3. The DC generator provides load to the PG by adjusting the value of resistance. The pitting fault was made by drilling three holes at the addendum of the sun gear, as shown in Figure 7. The transmission system is controlled by a PLC control cabinet. In this subsection, the PG with the pitting fault on the sun gear and the healthy PG were operated under the conditions shown in Table 4.

The vibration signal of the PG under each operating condition was collected by a vibration sensor, mounted on a ring housing. It is on the cross-section of the PG with sensitivity of 31.9 mv/(m/s^2^). The encoder was installed at the end of the drive motor shaft with resolution of 500 P/R. The experimental signals were collected by the YE6232 dynamic signal acquisition instrument. The sampling frequency was 71,428 Hz, and each signal length was 300,000 data points.

Figure 8 shows the spectrums of vibration signal of the healthy PG and the sun gear pitting fault PG, at 25% of the rated load and under three different speed conditions. As seen in Figure 8, the frequency spectrums of the vibration signal in the PG are complex, and the healthy planetary gearbox also displayed serious modulation phenomenon. It is difficult to find an obvious sideband in the spectrums. From the analysis of meshing frequency, the characteristic value of the healthy PG is lower than that of the sun gear pitting fault PG at 20% of its rated speed condition. However, the meshing frequencies are drowned out at 30% and 40% of its rated speed conditions. From the analysis of the third order harmonics of the meshing frequency, the characteristic values of the healthy PG are shown to be higher than those of sun gear pitting fault PG. Evidently, these phenomena cannot effectively show the existence of the faulty sun gear in the PG. Therefore, it is a challenge to successful incipient fault diagnosis in PGs. It is necessary to make the signal spectrum clearer and find a signature that can represent faults in a PG under different working conditions.

### 4.2. Data Analysis

The AOBS-FCER was then applied to analyze the measured vibration signals. Firstly, the raw signals were processed by the ADSA. In order to prevent the loss of useful information, the sampling frequency in the angular domain was set to 9000 sun gear Order. Secondly, the OB of each processed vibration signal was calculated. According to Section 2.1.1 and Table 3, the order spectrums of the main parameter to the sun gear are listed in Table 5. Thirdly, the AOBS was extracted. Since the internal components of a PG are inter-modulated, the range of modulation frequency was preliminarily set from 0 Order to 10 Order. The capability of the OB at different carrier frequency ranges is shown in Figure 9, and we can see that there is no obvious modulation coupled with the meshing frequency or its harmonics in the figure. In the low frequency range, the signal components of the mutually modulated signals are fewer than those in the high frequency range. As can be seen in Figure 9, the carrier frequency ranges from 10 Order to 30 Order, which met the requirements of selection. The optimal carrier frequency occurs at 25.29 Order, for the slice on the carrier frequency of 25.29 Order has the most intersections of multiple ridge lines. That denotes that the OB on 25.29 Order carries the most nonlinear coupling components. The AOBS under different rotation speeds is shown in Figure 10, Figure 11 and Figure 12, respectively. It can be observed from the three figures that the second, third and fourth sun gear orders can be identified on the AOBS, but their values are various under different sampling. The OB values of the second to fourth orders of sun gear order frequency are similar to those of the healthy PG under different conditions. However, the OB values of the second sun gear order are much higher than others for the sun gear pitting faulty PG.

The FCER is then used to analyse the modulation components on the AOBS, and the results are shown in Figure 13e. For comparative analysis, the second to fourth orders of sun gear order frequency are extracted and the results of OB are shown in Figure 13a,b and Figure 13c respectively. It is obvious that the second and third orders of sun gear order frequency extracted by the OB analysis can reflect the fault characteristic frequencies under the conditions of 30% and 40% rated speeds, but fail to represent the fault characteristic frequencies under the conditions of 20% rated speeds, inferred from Figure 13a,b. The fourth sun gear order frequency of the sun gear pitting fault PG was similar to that of the healthy PG (shown in Figure 13c) under the experimental conditions, so it cannot be used to identify the healthiness or the fault. The OSA-SER method was also applied to detect the weak fault features in the experimental data, and the results are shown in Figure 13d. The central frequency for the SER was chosen to be the 50.58 sun gear order frequency, which is the fifth order harmonic of meshing frequency, since the carrier frequency range from 40 to 60 sun gear order has more modulation components, as shown in Figure 9. However, this method failed to identify the weak fault features of the PG. By contrast, Figure 13e clearly shows that the AOBS-FCER can diagnose the weak fault in a PG under different conditions. Therefore, it proves that the proposed AOBS-FCER method is superior to the OB and the OAS-SER in the diagnosis of incipient faults in a PG.

## 5. Conclusions

A new method based on the AOBS and the FCER is proposed in this paper. This method can be applied to detect incipient faults in a PG with a floating sun gear. The AOBS method is proposed to extract the weak nonlinear components, and the FCER is developed to sum up and calculate the cognate features extracted by the AOBS. The contributions of this paper are highlighted as follows:(1)An optimized order bispectrum (OB) analysis, named AOBS, is proposed, which can extract modulations in low frequency areas.(2)A new statistical characteristic of FCER for the diagnosis of incipient faults in a PG is proposed. In comparison with the OB, the proposed FCER is superior in representing weak fault features.(3)Experimental analyses validate that the proposed AOBS-FCER method can solve the problems in feature extraction for incipient faults in PGs accurately and effectively.(4)In any further work, the effectiveness of the proposed method for detecting the incipient faults effecting the experimental system in real time will be investigated.

## Figures and Tables

**Figure 1 sensors-20-02433-f001:**
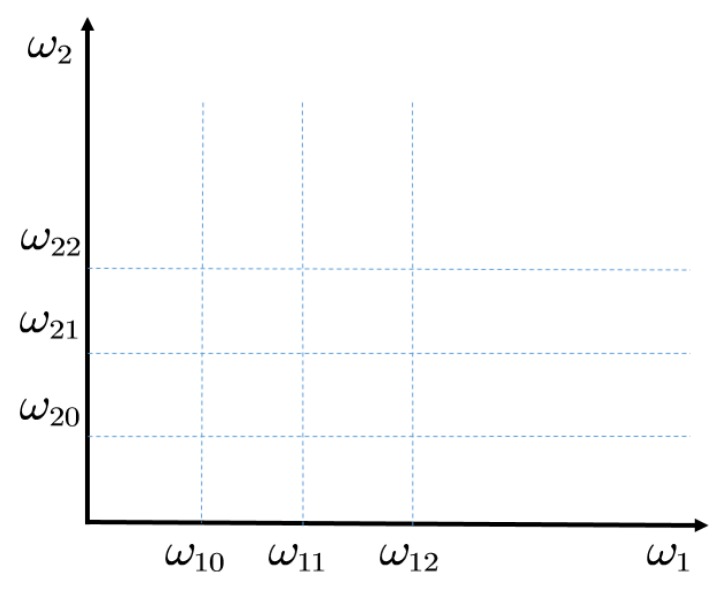
The formation of bispectral peaks on the carrier frequency and modulation frequency plane.

**Figure 2 sensors-20-02433-f002:**
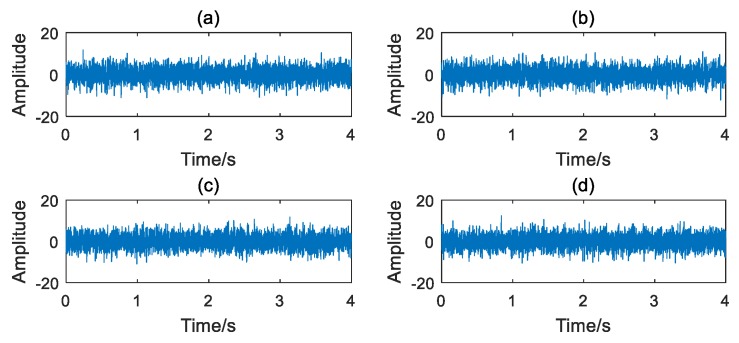
Waveforms of simulation signal: (**a**) Group a; (**b**) Group b; (**c**) Group c; (**d**) Group d.

**Figure 3 sensors-20-02433-f003:**
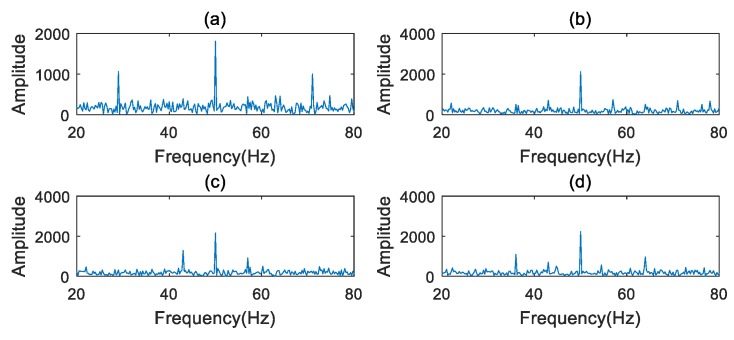
Simulation signals in frequency domain: (**a**) Group a; (**b**) Group b; (**c**) Group c; (**d**) Group d.

**Figure 4 sensors-20-02433-f004:**
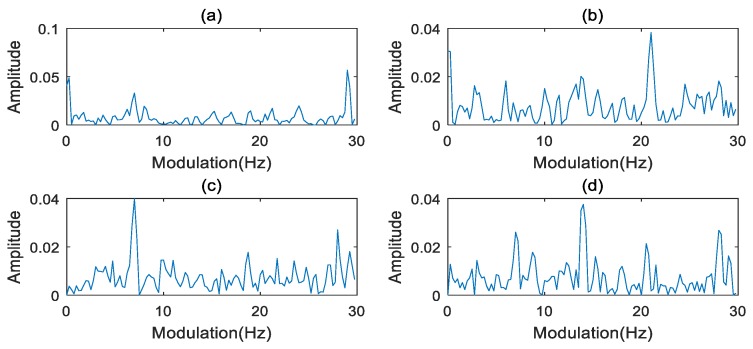
Modulated frequencies on the characteristic slice: (**a**) Group a; (**b**) Group b; (**c**) Group c; (**d**) Group d.

**Figure 5 sensors-20-02433-f005:**
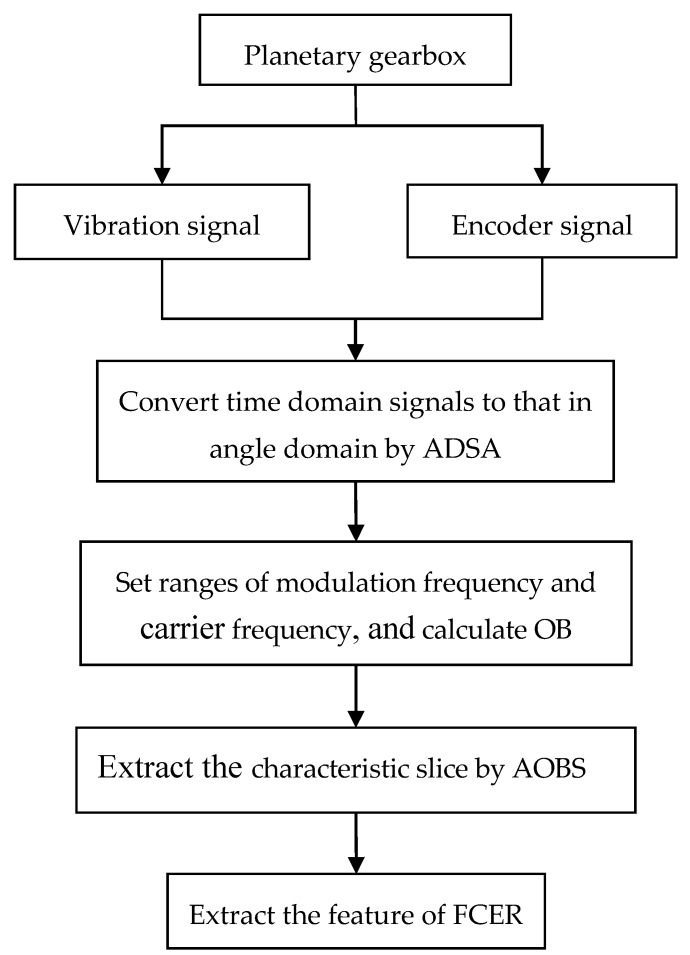
Flowchart of the proposed AOBS-FCER method.

**Figure 6 sensors-20-02433-f006:**
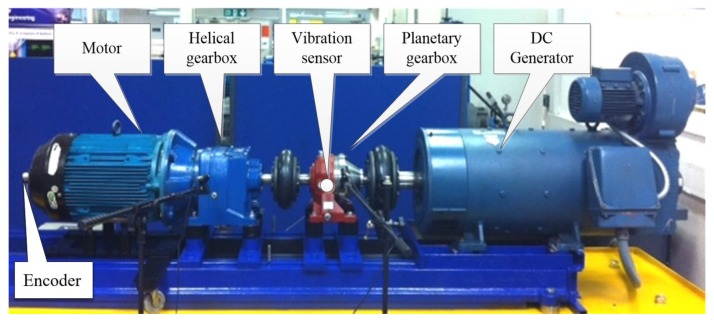
The planetary gearbox experiment system.

**Figure 7 sensors-20-02433-f007:**
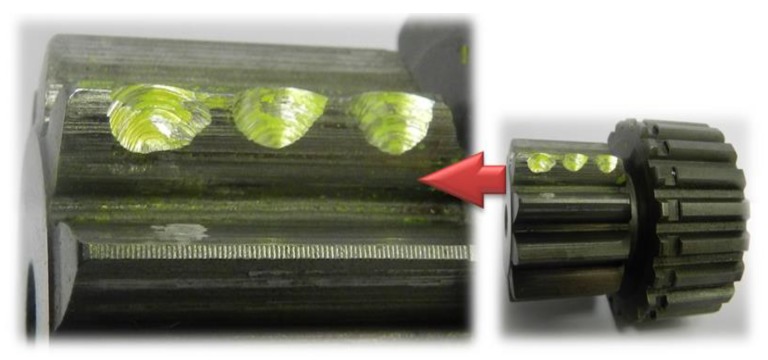
The pitting fault on sun gear.

**Figure 8 sensors-20-02433-f008:**
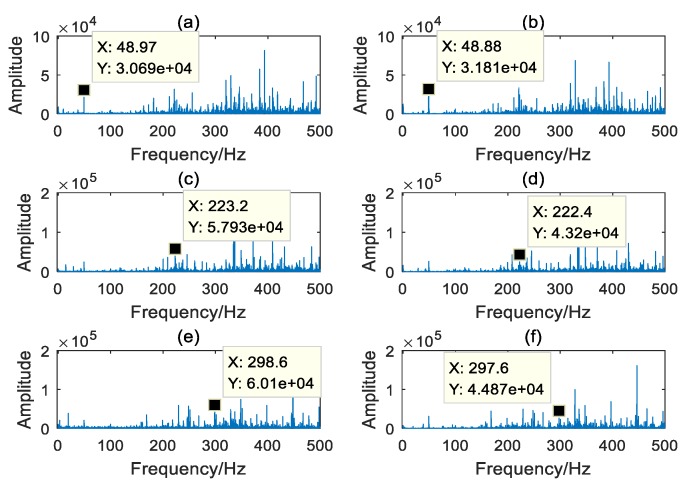
The spectrums of vibration signal under different speed conditions: (**a**) Healthy PG under 20% rated speed; (**b**) Sun gear pitting fault PG under 20% rated speed; (**c**) Healthy PG under 30% rated speed; (**d**) Sun gear pitting fault PG under 30% rated speed; (**e**) Healthy PG under 40% rated speed; (**f**) Sun gear pitting fault PG under 40% rated speed.

**Figure 9 sensors-20-02433-f009:**
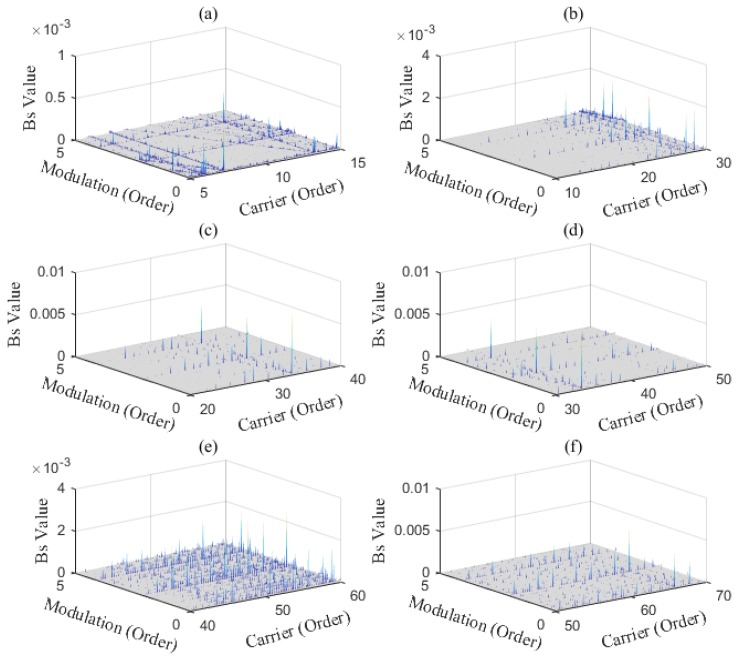
The OB values distribution under different carrier frequency ranges: (**a**) From 5 Order to 15 Order; (**b**) From 10 Order to 30 Order; (**c**) From 20 Order to 40 Order; (**d**) From 30 Order to 50 Order; (**e**) From 40 Order to 60 Order; (**f**) From 50 Order to 70 Order.

**Figure 10 sensors-20-02433-f010:**
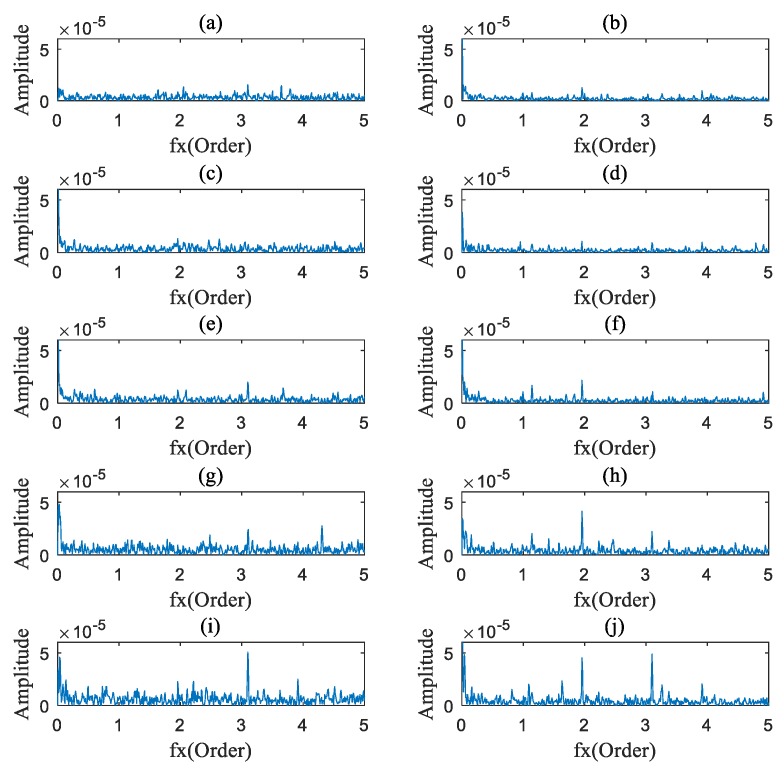
The characteristic slice under 20% rated speed conditions: (**a**) Healthy PG at 0% rated load; (**b**) Pitting fault on sun gear of PG at 0% rated load; (**c**) Healthy PG at 25% rated load; (**d**) Pitting fault on sun gear of PG at 25% rated load; (**e**) Healthy PG at 50% rated load; (**f**) Pitting fault on sun gear of PG at 50% rated load; (**g**) Healthy PG at 75% rated load; (**h**) Pitting fault on sun gear of PG at 75% rated load; (**i**) Healthy PG at 90% rated load; (**j**) Pitting fault on sun gear of PG at 90% rated load.

**Figure 11 sensors-20-02433-f011:**
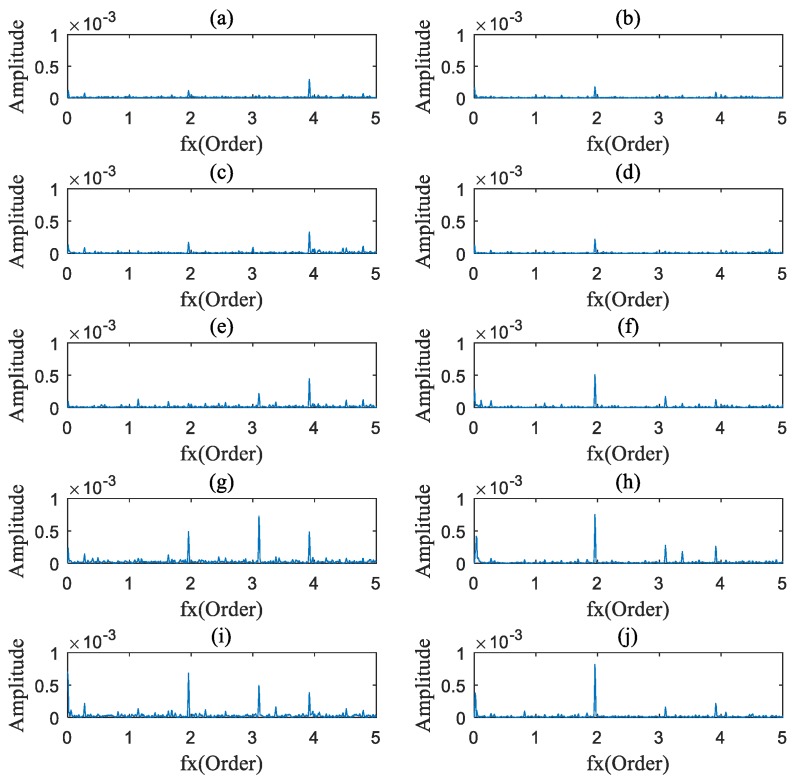
The characteristic slice under 30% rated speed conditions: (**a**) Healthy PG at 0% rated load; (**b**) Pitting fault on sun gear of PG at 0% rated load; (**c**) Healthy PG at 25% rated load; (**d**) Pitting fault on sun gear of PG at 25% rated load; (**e**) Healthy PG at 50% rated load; (**f**) Pitting fault on sun gear of PG at 50% rated load; (**g**) Healthy PG at 75% rated load; (**h**) Pitting fault on sun gear of PG at 75% rated load; (**i**) Healthy PG at 90% rated load; (**j**) Pitting fault on sun gear of PG at 90% rated load.

**Figure 12 sensors-20-02433-f012:**
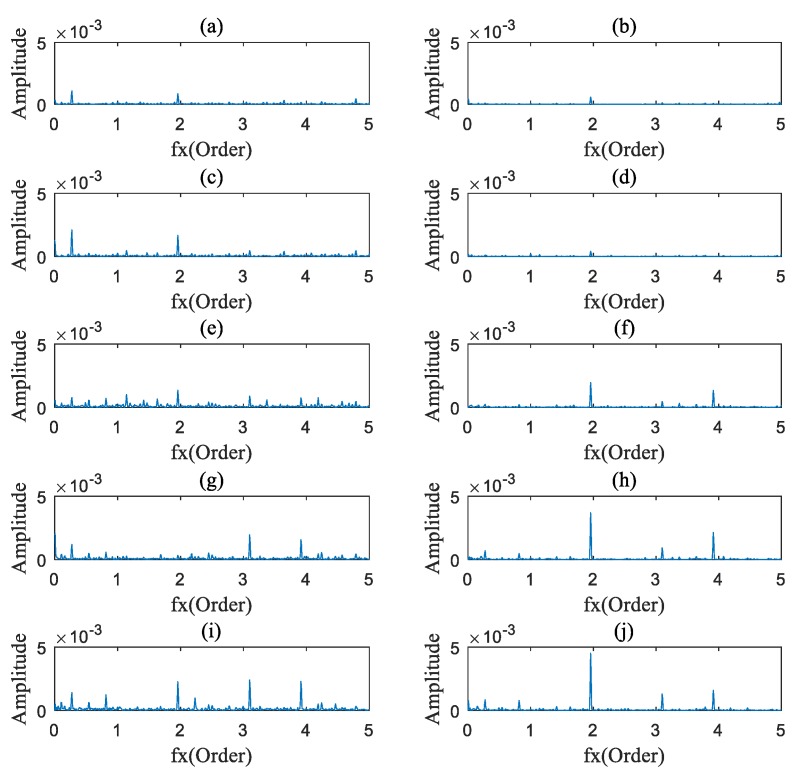
The characteristic slice under 40% rated speed conditions: (**a**) Healthy PG at 0% rated load; (**b**) Pitting fault on sun gear of PG at 0% rated load; (**c**) Healthy PG at 25% rated load; (**d**) Pitting fault on sun gear of PG at 25% rated load; (**e**) Healthy PG at 50% rated load; (**f**) Pitting fault on sun gear of PG at 50% rated load; (**g**) Healthy PG at 75% rated load; (**h**) Pitting fault on sun gear of PG at 75% rated load; (**i**) Healthy PG at 90% rated load; (**j**) Pitting fault on sun gear of PG at 90% rated load.

**Figure 13 sensors-20-02433-f013:**
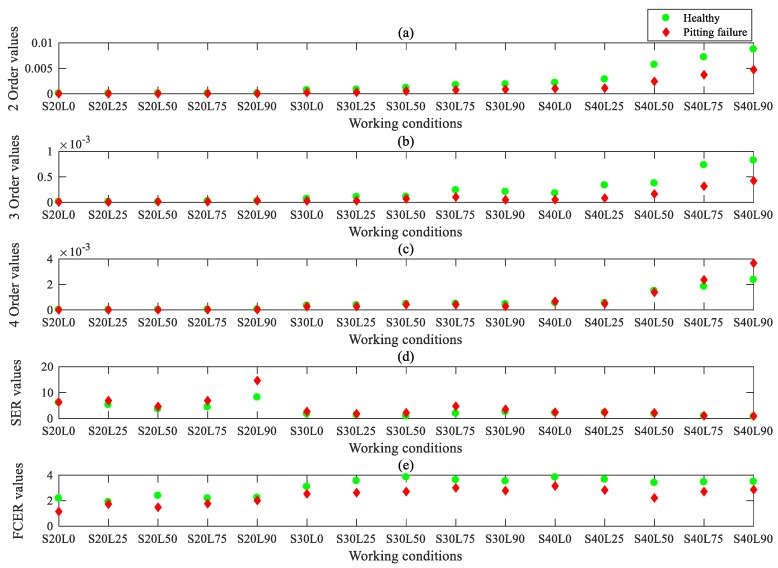
Feature extraction by different methods under different operating conditions: (**a**) Second sun gear order by order bispectrum; (**b**) Third sun gear order by order bispectrum; (**c**) Fourth sun gear order by order bispectrum; (**d**) SER values based on OSA; (**e**) FCER values based on AOBS.

**Table 1 sensors-20-02433-t001:** The weight of harmonics of the modulation frequency.

Group	1 Order	2 Order	3 Order	4 Order
a	0.25	0.15	0.5	0.1
b	0.3	0.22	0.28	0.2
c	0.6	0.05	0.18	0.17
d	0.1	0.5	0.2	0.2

**Table 2 sensors-20-02433-t002:** The results of SER and FCER for the simulation signal.

Group	SER	FCER
a	1.7930	4.5469
b	0.5725	4.3351
c	0.9924	4.3583
d	1.1901	4.3125

**Table 3 sensors-20-02433-t003:** Configuration parameters of the planetary gearbox.

Sun Gear	Planetary Gear	Inner Ring	Number of Planetary Gears	Transmission Ratio
10 teeth	24 teeth	62 teeth	3	7.2

**Table 4 sensors-20-02433-t004:** Operating conditions of the planetary gearbox system.

Speed	Load (Percent of Rated Load)				
20% rated speed	0	25%	50%	75%	90%
30% rated speed	0	25%	50%	75%	90%
40% rated speed	0	25%	50%	75%	90%

**Table 5 sensors-20-02433-t005:** Order spectrums of main parameters to the sun gear.

Sun Gear (Order)	Planet Gear (Order)	Carrier (Order)	Meshing (Order)
1	0.4167	0.1389	10

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
