# Peer review of "Fault Diagnosis of Planetary Gearbox Based on Adaptive Order Bispectrum Slice and Fault Characteristics Energy Ratio Analysis"

_sensors, 2020, doi:10.3390/s20082433_

Round 1

Reviewer 1 Report

Dear authors: The paper proposed a Planetary gearbox diagnosis method based on OB-FCER. The paper is well written. However, the paper is not innovative enough in theory and application. Therefore, the paper is not enough to be accepted, and the detailed comments are as follows. 1 Order bispectrum has been widely used in the field of fault diagnosis. 2 Vibration signal and Encoder signal are converted to the angle domain by ADSA, however, in the experimental part, the authors did not state the sampling frequencies of the two different signals. 3 In the analysis of experimental analysis results, the superiority of the proposed method cannot be directly demonstrated to readers. Firstly, we can't directly correspond to the diagnosis results according to the figure in the paper. Secondly, there is no comparison to reflect the advantages of the proposed method in diagnosis.

Author Response

Dear Reviewer,

      Many thanks for your comments. These comments are all valuable and very helpful for improving our manuscript as well as the important guiding significance to our researches. We have studied the comments carefully and have made the corrections which we hope to meet with your approval. It is true that the order bispectrum has widely used in the fault diagnosis of bearing and fixed-shaft gearbox, but its application in planetary gearbox is still to be exploited based on our literature review. In addition, we propose a new statistical feature of FCER based on the feature extraction of an adaptive order bispectrum slice (AOBS). Maybe it is because our manuscript does not clearly highlight the innovation points, so we modified it,and the revised parts are marked in red in the manuscript.  Please see the attachment.

Reviewer 2 Report

This paper presented very interesting method of planetary gearbox fault diagnosis. It can be improved:

  • at section 4.1. Experimental setup: presentation of basic technical data / specifications of vibration sensor and encoder, sensor mounting method; explanation whether the spectrum of vibration signal applies to vibration in one selected axis or refers to the components of spatial vibration in the xyz axis,
  • Figure 6: improving the readability of charts by enlarging them and using the same scale on the Bs Value axis,
  • Figure 7, 8, 9: using the same scale amplitude axis at least within the results of the same PG load (within each of the drawings) - this will allow easier graphical comparison of the resuts with each other,
  • some minor editorial errors: line 222 - beginning of a capital letter; line 196 and line 199 - missing space; line 305 - double space.

Reviewer 3 Report

The authors wrote a sounding paper. Some additional details should be included in the final manuscript, see my points below. Besides those points, the introduction section should be improved to provide sufficient background to the studied problem and list all relevant references. Moreover, the research design and the experimental methodology should be improved. Certain steps in the experimental methodology are missing. The authors should list all steps and technical details for the sake of test reproducibility. Finally, the results should be presented more meaningfully, more to the point and more profoundly, without unnecessary plots. The conclusion section should be more closely tightened with the findings and novelty. 

Specific comments:

-model-based approaches could be as well researched in the introduction section
- the main contributions are not clearly stated at the end of the introduction section
- in Eq. (2), how is the equal angle interval calculated?
- the authors should provide the model of incipient faults. Do you have an analytical model? Can you derive an empirical model from the experimental results?
- in the proposed method, it is not clear when (time) a fault is declared
- implementation aspects should be more thoroughly discussed
- how does the proposed method compare to conventional signal-based fault diagnosis? What are the pros/cons of the proposed method?

Reviewer 4 Report

GENERAL COMMENTS

This paper deals with the problem of Fault Diagnosis of Planetary Gearbox Based on Order Bispectrum and Fault Characteristics Energy Ratio. Authors try to combine the advantages of these known tools.

REMARKS AND QUESTIONS:

- The introduction needs to be rewritten to sort out largely the existing fault diagnosis methods and point out the corresponding features.

- No originality in this paper and the proposed approach is reduced to two pages only (pages 3 and 4).

- The results show the impact of the considered fault on considered features (OB and FCER). It is normal that we do not have the same results. I cannot conclude about the performances of the proposed approach only by comparing healthy and faulty modes for several working conditions. It is more interesting that authors show the results for incipient faults in progress on the system (in your case for example, one hole, two holes, three …. and so on).

- It is interesting also to discuss the question of applicability of this approach on real time operating.

Round 2

Reviewer 1 Report

I am fine with this version.

Author Response

    Many thanks again for your comments. These comments are all valuable and very helpful for improving our manuscript as well as the important guiding significance to our researches. We have studied the comments carefully and have made the corrections which we hope to meet with your approval. The revised parts are detailed in the attachment. Please see the attachment.

Reviewer 3 Report

All my comments have been responded. The authors could further improve the introduction and conclusions sections as well as the overall flow/structure of the paper. The description of the experimental methodology could be more clear. Some figures could be presented in more compact and meaningful way.

Author Response

    Many thanks again for your comments in the paper. These comments are all valuable and very helpful for improving our manuscript as well as the important guiding significance to our researches. We have studied the comments carefully and have made the corrections which we hope to meet with your approval. The revised parts are detailed in the attachment. Please see the attachment.

Reviewer 4 Report

Thank you for the responds.

  • The introduction needs to be rewritten to sort out largely the existing fault diagnosis methods and point out the corresponding feature.

OK

  • No originality in this paper and the proposed approach is reduced to two pages only (pages 3 and 4).

The content is not sufficient. 

  • The results show the impact of the considered fault on considered features (OB and FCER). It is normal that we do not have the same results. I cannot conclude about the performances of the proposed approach only by comparing healthy and faulty modes for several working conditions. It is more interesting that authors show the results for incipient faults in progress on the system (in your case for example, one hole, two holes, three …. and so on).

Lack of relevance in the analysis of results

  • It is interesting also to discuss the question of applicability of this approach on real time operating.

Suggested for future work by the authors

Author Response

(The authors gave the same response as above.)
